# Management and Oncologic Outcomes of Close and Positive Margins after Transoral CO_2_ Laser Microsurgery for Early Glottic Carcinoma

**DOI:** 10.3390/cancers15051490

**Published:** 2023-02-27

**Authors:** Cinzia Mariani, Filippo Carta, Mauro Bontempi, Valeria Marrosu, Melania Tatti, Valeria Pinto, Clara Gerosa, Roberto Puxeddu

**Affiliations:** 1Unit of Otorhinolaryngology, Department of Surgery, Azienda Ospedaliero-Universitaria di Cagliari, University of Cagliari, 09100 Cagliari, Italy; 2Unit of Pathology, Department of Medical Sciences and Public Health, Azienda Ospedaliero-Universitaria di Cagliari, University of Cagliari, 09100 Cagliari, Italy; 3ENT Department, King’s College Hospital London-Dubai, Dubai P.O. Box 340901, United Arab Emirates

**Keywords:** laryngology, early glottic cancer, laser surgery, margins, outcomes

## Abstract

**Simple Summary:**

The management of close and/or positive margins after transoral CO_2_ laser microsurgery (CO_2_ TOLMS) is still an ongoing matter of discussion. Different options have been suggested on the basis of the number of the involved margins (single vs. multiple) and site (deep vs. superficial): strict follow-up, revision surgery or radiotherapy. However, precise indications about additional treatment and its effective impact on local control and survival rates are still lacking. The authors reviewed 351 patients with early glottic cancer treated using CO_2_ TOLMS with the aim of analyzing the impact of margin status on local control and survival, and discussing the therapeutic options in cases of close and positive margins.

**Abstract:**

Background: The present study analyzed the impact of margin status on local control and survival, and the management of close/positive margins after transoral CO_2_ laser microsurgery for early glottic carcinoma. Methods: 351 patients (328 males, 23 females, mean age 65.6 years) underwent surgery. We identified the following margin statuses: negative, close superficial (CS), close deep (CD), positive single superficial (SS), positive multiple superficial (MS), and positive deep (DEEP). Results: A total of 286 patients (81.5%) had negative margins, 23 (6.5%) had close margins (8 CS, 15 CD) and 42 (12%) had positive margins (16 SS, 9 MS, 17 DEEP). Among the 65 patients with close/positive margins, 44 patients underwent enlargement, 6 radiotherapy and 15 follow-up. Twenty-two patients (6.3%) recurred. Patients with DEEP or CD margins showed a higher risk of recurrence (hazard ratios of 2.863 and 2.537, respectively), compared to patients with negative margins. Local control with laser alone, overall laryngeal preservation and disease-specific survival decreased significantly in patients with DEEP margins (57.5%, 86.9% and 92.9%, *p* < 0.05). Conclusions: Patients with CS or SS margins could be safely submitted to follow-up. In the case of CD and MS margins, any additional treatment should be discussed with the patient. In the case of DEEP margin, additional treatment is always recommended.

## 1. Introduction

Transoral CO_2_ laser microsurgery (CO_2_ TOLMS) [1] represents a well-standardized and minimally invasive approach allowing good oncological and functional results in patients with early glottic squamous cell carcinoma (SCC) [2,3]. This ultraconservative approach can be burdened by an increased incidence of close and positive margins of resection, which are unanimously associated with a higher risk of local relapse [4,5,6,7,8].

In the case of inadequate margins, different options have been suggested on the basis of the number of the involved margins (single vs. multiple) and site (deep vs. superficial): strict follow-up, revision surgery or radiotherapy (RT) [9]. However, precise indications for additional treatment and its effective impact on local control and survival rates are still debated.

There is a common consensus that further treatments are required in patients with positive deep margin of resection at definitive histology, while the management of close and/or positive superficial margins is still an ongoing matter of discussion [9] since additional resection can potentially hinder the functional results.

The high rate of false positive margins after enlargement reported in the literature (up to 80%) [6,9,10,11,12,13] has driven several authors to reduce second-look procedures in recent years, implementing a watchful waiting policy [6,9].

We reviewed a large homogeneous cohort of patients affected by Tis-T1-T2 glottic SCC treated using CO_2_ TOLMS with the aim of analyzing the impact of margin status on local control, survival and organ preservation. The decision-making process and effective indication of additional therapeutic options in cases of close and positive margins have been discussed.

## 2. Materials and Methods

The authors performed a retrospective analysis of 351 patients with early glottic SCC (Tis-T1-T2) treated by the senior author with CO_2_ TOLMS from October 1993 to November 2005 and from December 2010 to December 2020 at the Department of Otorhinolaryngology of an Italian institution (ethics committee protocol number 895/2018). All the patients included in the study had no clinical involvement of the lymph node at the time of surgery.

During the preoperative work up, all patients underwent fiberlaryngoscopy, while computed tomography (CT) or magnetic resonance imaging (MRI) of the neck with contrast medium were considered necessary in selected patients to rule out invasion of the paraglottic space (PGS), of the preepiglottic space and of the cartilage. Intraoperative work up was always performed using rigid 0° and 70° scopes. From the end of 2013, preoperative and intraoperative endoscopic work up was coupled with narrow-band imaging (NBI) (Olympus Medical Systems Corporation, Tokyo, Japan) and the IMAGE1 S System^TM^ (Storz, Tuttlingen, Germany) plus enhanced contact endoscopy (ECE) [14].

On the day of surgery, all patients received ceftriaxone (1000 mg IV) or, as a substitute if allergic, ciprofloxacin (400 mg IV), according to the antibiotic prophylaxis protocol of our institution.

All patients underwent CO_2_ TOLMS under general anesthesia with orotracheal intubation (Mallinckrodt laser safe tube, I.D. 5.0–7.0 mm; Athlone, Ireland). Sharplan 1030 and Acupulse CO_2_ lasers with an Acuspot, Acublade 712 micromanipulator and Digital AcuBlade^TM^ (Lumenis^®^, Yokneam, Israel) set on the superpulsed mode (10 W, continuous, acublade 1–3 mm) were used in most of the cases. The UltraPulse/Surgitouch CO_2_ laser (Lumenis^®^, Yokneam, Israel) was used from 2020.

Adequate laryngeal exposure in microlaryngoscopy was obtained using the Kleinsasser laser laryngoscopes modified by Rudert with the Riecker–Kleinsasser suspension system (Karl Storz, Tüttlingen, Germany).

Endoscopic cordectomies were classified according to the European Laryngological Society (ELS) [15,16].

Resections were always performed using an en bloc procedure when the volume of the tumor allowed it. Larger tumors were removed using a piecemeal technique. When indicated, the anterior commissure was resected through a subperichondrial dissection. Resections were performed in macroscopic free margins. Specimens were sent for histology opportunely oriented by the surgeon by staining the superior edge with ink to obtain the precise mapping of the lesions, also after piecemeal removal. Frozen sections were not routinely performed because they are unrepresentative of the whole mucosal margins, and the time of execution can become excessive for organizational reasons and hospital logistics.

After definitive histology, all lesions were staged and restaged according to the eighth edition of the Union for International Cancer Control–American Joint Committee on Cancer (UICC–AJCC) TNM staging system [17].

According to Fiz et al. [5], the margin status was classified as follows: negative, close (tumor–margin distance < 1 mm) superficial (CS), close deep (CD), positive (presence of at least carcinoma in situ at the surgical margin) single superficial (SS), positive multiple superficial (MS) and positive deep (DEEP).

In the case of close or positive margins, intraoperative recording was reviewed and discussed in a multidisciplinary team. The policy after histology was as follows: CO_2_ transoral enlargement or postoperative RT was always performed in the case of DEEP or MS margins; SS and CD margins were almost constantly enlarged with a laser, except selected cases who underwent a close wait-and-see policy; CS margins were managed with close follow-up.

Patients scheduled for a second look with CO_2_ TOLMS were treated at 30 to 40 days after the first cordectomy, because by that point the scar tissue is completely healed and the glottic aspect can be better evaluated.

Voice rehabilitation and regular follow-up were scheduled according to the NCCN guidelines in all cases [18]: patients underwent fiberlaryngoscopy every month during the first year, every 2 months during the second year and every 3–4 months until the fifth year after surgery, in the absence of any recurrence and/or secondary disease. From 2013, fiberlaryngoscopy was coupled with NBI. Patients included in the present study were followed up from the date of surgery until December 2022, when possible.

Statistical analysis was performed on the basis of the data reported as Appendix A, using GraphPad Prism software (GraphPad, San Diego, CA, USA). Survival probabilities over time were estimated using the Kaplan–Meier method, considering the six different types of margins (negative, CS, CD, SS, MS, DEEP). The entry point was the date of laser cordectomy. The first studied outcome was disease-specific survival (DSS), with the end point being patient’s death due to laryngeal cancer or last follow-up. The second outcome was recurrence-free survival (RFS), with the end point set at the date of recurrence or at the last available visit. The third outcome was local control with laser alone (LCL), with the end point set at the date of RT or open procedure for recurrences. Organ laryngeal preservation (OLP) was the fourth measured outcome, with the end point set at the date of total laryngectomy or at last follow-up. The log-rank (Mantel–Cox) test was applied to compare recurrence rates between patients with negative margins versus patients with close/positive margins. A *p* value < 0.05 was considered to be statistically significant.

The influence of the routine intraoperative use of NBI, IMAGE1 S and ECE in the incidence of positive superficial margins was evaluated. 

## 3. Results

Three hundred and fifty-one patients (328 males, 23 females, mean age 65.6 years, age range 29–90 years) with early glottic SCC treated with CO_2_ TOLMS were included in the study. Of the 351 patients treated with CO_2_ TOLMS, 34 (9.7%) underwent type I cordectomy, 94 (26.8%) type II cordectomy, 77 (21.9%) type III cordectomy, 21 (6%) type IV cordectomy, 122 (34.8%) type V cordectomy and 3 (0.8%) type VI cordectomy. Patient and tumor characteristics, and the numbers and types of surgical cordectomies are detailed in Table 1.

The mean hospitalization time was 2.9 days. Three patients experienced a postoperative bleeding that required an endoscopic cautery under general anesthesia. Thirty-four patients (9.7%) developed an anterior glottic web. Among them, the 12 patients with moderate to severe symptoms were managed endoscopically with the laser incision of the web and the harvesting of a mucosal microflap, while the 22 patients with mild symptoms were referred to voice therapists, as suggested in the literature [19].

Two hundred and eighty-six patients (81.5%) had negative margins after primary surgery, while 65 patients (18.5%) had close or positive margins. Twenty-three patients (6.5%) had close margins, among whom 8 had CS and 15 had CD margins. Forty-two patients (12%) had positive margins: 16 had SS margin, 9 MS margins and 17 DEEP margin (Table 1).

Seventeen out of the 229 patients (7.4%) who underwent CO_2_ TOLMS before the systematic intraoperative use of enhancement systems (NBI, IMAGE1 S and ECE) had positive superficial margins. After the implementation of bioendoscopy, only 8 out of 122 patients (6.6%) experienced positive superficial margins. Therefore, the use of enhancement tools reduced the incidence of positive superficial margins, although the decrease was not statically significant (*p* = 0.76).

Forty-four (67.7%) of the 65 patients with close/positive margins underwent CO_2_ laser enlargement, obtaining negative margins in all cases. Among these patients, 12 had DEEP margin, 8 had MS margins, 14 had SS margin and 10 had CD margin. Definitive histology showed the presence of residual carcinoma in only 8 (18.2%) of these 44 cases: 3 patients with initially DEEP margin, one with MS margins, and 4 with SS margin. Definitively, 330 patients (94%) showed negative margins after primary surgery and subsequent enlargements.

A total of 7 patients underwent postoperative RT: six (9.2%) of the 65 patients had close/positive margins (5 with DEEP margin and one with MS margins), and one patient had negative margins but lymphovascular invasion at definitive histology.

Fifteen (23.1%) of the 65 patients with close/positive margins underwent a close wait-and-see policy, among whom 8 had CS margin, 5 had CD margin and 2 had SS margin.

The mean follow-up was 4.97 years.

Twenty-two of the 351 patients (6.3%) experienced recurrence: laryngeal recurrence in 21 cases and nodal recurrence in 1 case. No patients developed distant metastasis. Recurrence occurred in 17 patients with negative margins, in 3 patients with DEEP margin who underwent surgical enlargement without evidence of residual tumor at histology, and in 2 patients with CD margin who underwent follow-up. Twenty-one patients underwent salvage treatment; one patient refused additional treatment and died of disease. Salvage therapy of the 21 patients with recurrences included: CO_2_ TOLMS alone in 11 cases, CO_2_ TOLMS and RT in 1 case, RT alone in 1 case, type II OPHL in 1 case, type III OPHL in 1 case, type II OPHL and RT in 1 case, total laryngectomy (TL) in 3 cases, TL and RT in 1 case and radical neck dissection and RT in 1 case (Table 2). 

The five-year DSS, RFS, LCL and OLP of the whole series were 99.6%, 92.9%, 94.6% and 98.2%, respectively. Survival rates and Kaplan–Meier survival curves relative to the different subtypes of margins are reported in Table 3 and Figure 1.

In the univariate analysis, patients with DEEP margin showed a 2.863 (*p* = 0.08) times increased risk for recurrence compared to patients with negative margins (Table 4). Furthermore, in the case of DEEP margin, both LCL, OLP and DSS decreased in a statistically significant way: 57.5%, 86.9% and 92.9%, respectively, in patients with DEEP margin versus 96.4%, 98.7% and 100%, respectively, in patients with negative margins, *p* < 0.05. Patients with CD margin experienced a 2.537 (*p* = 0.2) times increased risk of recurrence (RFS of 86.7% versus 93.2% in patients with negative margins) (Table 4).

## 4. Discussion

The surgical margins required in early glottic SCC are narrower than those considered necessary in other head and neck cancers because of the scarce glottic submucosal lymphatic network. In the literature, margins ≥ 1 mm of healthy tissue are generally considered adequate in patients treated with CO_2_ TOLMS [4,20,21]. The high magnification available with the operative microscope and modern biologic endoscopic techniques makes the surgical approach with such narrow margins easier. In our series, we found negative margins in 286 cases (81.5%) and close margins in 23 cases (6.5%), whereas positive margins were present in 42 cases (12%). Our rate of positive margins is, encouragingly, in the range reported in the literature (9.3–45.4%) [4,5,6,7,10,22,23,24,25,26].

Different prognostic factors, such as stage or anterior commissure involvement, may be associated with local recurrence, but various studies have demonstrated that positive margins represent an independent risk factor for local failure [9,10,21,25,27]. In the literature, it is reported that a local recurrence rate ranging from 3.1% to 22.8% is observed in cases of negative surgical margins, while the recurrence rate rises from 8% to 51% in cases of positive margins [10]. This great variability of incidences of local relapse could be related to the experience of the surgeon and to the different interpretation of the margin status performed by the pathologist. According to Fiz et al. [5], we analyzed our oncological outcomes, classifying the margins as negative, CS, CD, SS, MS and DEEP.

The intraoperative use of rigid endoscopy with bio endoscopic tools such as NBI and IMAGE1 S has been suggested as a useful tool in achieving optimal superficial margins outlines, and has been shown to potentially decrease the rate of positive superficial margins [5,14,28,29,30]. According to the literature, in our series, the systematic use of NBI, IMAGE1 S and ECE decreased the number of positive superficial margins (6.6% vs. 7.4%), although the difference was not statistically significant (*p* = 0.76).

In the present series, the CS and SS margins did not negatively impact RFS. Close superficial margins were always managed with close follow-up. Regarding SS margin, in the early years of the present series, our policy was for systematic enlargement, but since the majority of revision surgeries resulted in negative specimens (71.4%), in the second part of the present series, we adopted a strict endoscopic follow-up, performing a second CO_2_ TOLMS only when the surgeon expressed doubts concerning the resection, as suggested in the literature [6,21,31,32,33]. Avoiding unnecessary surgical enlargements has the advantage of sparing the voice quality, because a second TOLMS clearly results in a further loss of tissue, increasing the scarring of the residual vocal cords. The low incidence of cases with residual disease at histology after the second look could be explained by the loss of the narrow healthy tissue due to the laser effect (thermal damage), and/or because of the shrinkage of the specimen [6,9,12]. Human tissues, especially the mucosa, have an intrinsic propensity to undergo shrinkage after surgical resection because of the presence of contractile proteins in the connective tissue and their release from the surrounding structures. In the literature, a mean shrinkage of mucosal specimens after CO_2_ TOLMS, from intralaryngeal measurement to postresection, of 3.8 ± 0.3 mm in the anteroposterior length of the glottic plane is reported [34]. Such important shrinkage could explain the high number of unnecessary enlargements to obtain wider or free margins. Therefore, careful harvesting and orientation of the surgical specimen is mandatory for the correct evaluation of the tumor extent. Several protocols have been described in the literature to improve margin assessment. Michel et al. [35] and Aluffi Valletti et al. [6] suggested the use of two different colored inks to tag superficial and deep mucosal sides before formalin fixation. In the present series, all the specimens were systematically three-dimensionally oriented by staining the superior margin with ink, and were analyzed by a dedicated pathologist. 

Some authors reported an increased risk of recurrence in the presence of MS margins [5,36]. In our series, none of the nine cases with MS margins experienced recurrences, and postoperative RT was deemed necessary in one patient with multifocal carcinoma. In patients with multifocal SCC, it is difficult to assess the true superficial extension of the lesion and obtain a radical excision; thus, multiple CO_2_ TOLMS are needed, and, in some instances, RT represents an additional tool. In the eight patients with MS margins who underwent CO_2_ laser enlargement, the resection was carried out starting from a wider macroscopically free margin including the scar of the previous surgery. Definitive histology showed the presence of residual carcinoma in only one (12.5%) of these eight patients. These data suggest that the majority of patients with MS margins could be overtreated with revision surgery. 

Nowadays, the use of HD flexible endoscopy with bioendoscopy performed in the office setting during the follow-up improves the accuracy of the early detection of persistent or recurrent disease after CO_2_ TOLMS. This could be a reason to shift to a less aggressive attitude concerning second-look procedures in cases of doubtful or positive superficial margins, even multiple, opting for a close follow-up with NBI endoscopy [29].

Fiz et al. [5] reported that close deep margins were related to an increased number of relapses, with an RFS of 77.1%. This was also confirmed by our findings, since patients with CD margin experienced a lower RFS (86.7% versus 93.2% in patients with negative margins) with a 2.537 times increased risk of recurrence; however, this result was not statistically significant (*p* = 0.2). Two of the five patients with CD margin who underwent follow-up experienced recurrence and were managed with additional CO_2_ TOLMS. Among the 10 patients with CD margin who underwent the second look, none showed residual carcinoma at histology. Ultimately, LCL, OLP and DSS were not negatively impacted by the CD margin status. As a consequence, we believe that the CD margin does not represent an absolute indication for additional treatment, although a strict follow-up with imaging is mandatory to detect early recurrence.

According to the literature, DEEP margins are generally associated with the highest recurrence rate [5,6,7,8]. In our experience, patients with DEEP margin showed an increased risk of recurrence compared to patients with negative margins, with a hazard ratio of 2.863 (*p* = 0.08). Five patients required RT and one patient underwent total laryngectomy after recurrence, whereas one patient refused treatment after recurrence and died of disease. Consequently, both LCL, OLP and DSS were negatively affected in patients with DEEP margin (respectively 57.5%, 86.9% and 92.9% versus 96.4%, 98.7% and 100% in patients with negative margins, *p* < 0.05). 

In our series, none of the five patients who underwent postoperative RT after DEEP margin at histology experienced recurrence. In the case of positive margins, the role of postoperative RT is still debated. Some authors found a benefit in RFS after postoperative RT, whereas others could not demonstrate any significant difference in patients submitted for adjuvant treatment when compared with those followed up with a compulsory protocol of surveillance [6,9,36]. Furthermore, postoperative RT results in a multimodal therapeutic approach for early tumors that could have been managed by RT alone from the beginning, with additional biological and economic costs [20]. Moreover, the patient would lose the possibility of being treated with RT in the event of a laryngeal second primary or recurrence [20]. 

A recent review [37] pointed out that the application of postoperative RT in patients with positive margins following the first resection depends on the confidence of the surgeon with wider resection, and can range from 10% [36,38] to 44% [4,39,40]. However, if a second CO_2_ TOLMS is judged unlikely to result in “true negative margins”, then open partial surgery or RT are the most appropriate choices [37]. 

Among the 12 patients with DEEP margin who underwent additional CO_2_ TOLMS, 3 patients recurred, despite surgical enlargement with no residual tumor at the histology. These “false” negative margins are difficult to interpret: although missing at histology, the residual disease cannot be excluded. In two of these cases, the patients were initially treated with extended type V cordectomies for cT2 tumors, with clinical involvement of the posterior third of the vocal cord. In these cases, histology confirmed the presence of the cancer immediately lateral to the vocal process and close to the PGS. A lower local control can be observed in the case of understaging of T2/T3 tumors. The misdiagnosis of cT2 can be associated with the complexity of the assessment of the PGS. The correct identification of the involvement of the posterior PGS is essential to choose the appropriate therapeutic strategy, since patients with posterior glottic tumors have poor local control when treated with CO_2_ TOLMS [41]. The limit that separates the anterior from the posterior laryngeal compartments is a virtual plane described as tangential to the vocal process and perpendicular to the ipsilateral thyroid lamina. We believe that, in the case of posterior DEEP margins after extended cordectomies, the surgeon should consider the possibility of micro infiltration of the posterior PGS and, consequently, decide on postoperative RT or open surgery. 

Evidently, our analysis was limited by the reduced number of each subgroup of patients, which did not allow us to perform a multivariate analysis. Therefore, to draw any definite conclusions, large controlled multi-center retrospective and prospective trials are needed.

## 5. Conclusions

The present study confirms that, although CO_2_ TOLMS in early glottic cancer offers optimal oncological outcomes, the margin status impacts on local control, and it is mandatory to stratify the different types of margins for their distinctive prognostic significance.

In the present series, the CS and SS margins behaved similarly, and could be followed-up after adequate counseling with the patients.

We observed that the CD and MS margins do not have a statistically significant negative impact on DSS, RFS, LCL or OLP, and any additional treatment should be thoroughly discussed with the patient so as to avoid unnecessary overtreatment.

Patients with DEEP margin should always undergo additional treatment, and if a second CO_2_ TOLMS is judged unlikely to result in “true negative margins”, open surgery or RT must be considered.

## Figures and Tables

**Figure 1 cancers-15-01490-f001:**
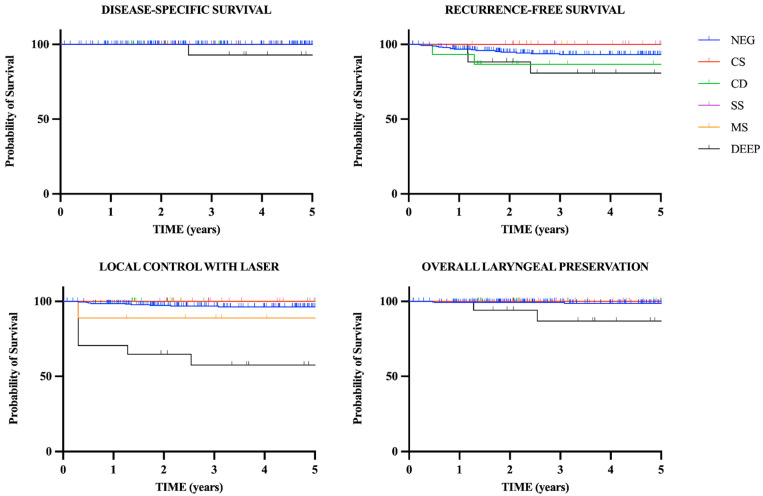
Kaplan–Meier curves for the entire cohort of patients showing disease-specific survival, recurrence-free survival, local control with laser alone and overall laryngeal preservation in relation to margin status.

**Table 1 cancers-15-01490-t001:** Patients who underwent CO_2_ TOLMS for early glottic cancer.

Variables	No. of Patients
All	351
Age	65.6 (range 29–90 years)
Male/female	328/23
Type of surgical cordectomies	34 type I cordectomy 94 type II cordectomy 77 type III cordectomy 21 type IV cordectomy 122 type V cordectomy (61 a, 7 ab, 4 abc, 11 abcd, 2 abd, 10 ac, 8 acd, 7 ad, 7 b, 3 bc, 2 c) 3 type VI cordectomy
Clinical N classification	351 cN0
Pathological T classification	34 pTis 193 pT1a 61 pT1b 63 pT2
Margin status	286 NEG 8 CS 15 CD 16 SS 9 MS 17 DEEP

NEG = negative. CS = close superficial. CD = close deep. SS = positive single superficial. MS = positive multiple superficial. DEEP = positive deep. a = extended cordectomy encompassing the contralateral vocal fold. b = extended cordectomy encompassing the arytenoid. c = extended cordectomy encompassing the ventricular fold. d = extended cordectomy encompassing the subglottis.

**Table 2 cancers-15-01490-t002:** Recurrences after CO_2_ TOLMS for early glottic cancer.

Patient/Sex/Age (Years)	Type of Cordectomy	pT	Margin Status after Primary Treatment	CO_2_ Laser Enlargement	Site of Relapse/Time of Relapse (Years)	Salvage Treatment	Outcome/Time of Last Follow-Up (Years)
DG/M/76.3	Va	1b	NEG	-	Larynx/1.7	CO_2_ TOLMS and radiotherapy	DOC/2.5
FE/M/68.5	IV	1b	NEG	-	Larynx/0.3	CO_2_ TOLMS	NED/7
MG/M/70.1	II	1a	NEG	-	Larynx/0.6	Type II horizontal laryngectomy and radiotherapy	NED/5
VA/M/61.9	II	1a	NEG	-	Larynx/2.4	CO_2_ TOLMS	DOC/11.1
RB/M/82.7	Vabc	2	DEEP	Yes (no residual tumor at histology)	Larynx/2.4	Refused treatment	DOD/2.5
DE/M/72.8	II	1a	NEG	-	Larynx/0.7	CO_2_ TOLMS	DOC/4.9
LS/M/76.6	Va	2	NEG	-	Larynx/1.7	CO_2_ TOLMS	DOC/4.3
FG/M/64.1	Vacd	2	NEG	-	Larynx/1.3	Type II horizontal laryngectomy	NED/11
MS/M/53.5	IV	1a	NEG	-	Larynx/3	Total laryngectomy	NED/7.8
SC/M/61.4	III	1a	NEG	-	Larynx/0.8	CO_2_ TOLMS	NED/6.6
CS/M/60.9	II	1a	NEG	-	Larynx/1.8	CO_2_ TOLMS	NED/2.1
ZF/M/71.5	Vabcd	2	NEG	-	Larynx/0.2	Total laryngectomy	DOC/0.5
PB/M/64.1	Vac	2	DEEP	Yes (no residual tumor at histology)	Larynx/1.2	Total laryngectomy	NED/5.2
RM/M/73.8	I	1a	NEG	-	Larynx/1.2	Radiotherapy	NED/5.7
DS/F/62.8	Vabcd	2	NEG	-	Larynx/0.5	Total laryngectomy and radiotherapy	NED/5.1
BM/M/51.7	Vb	1a	NEG	-	Larynx/2.1	Type III horizontal laryngectomy	NED/3.2
SG/M/68.3	Vac	1b	NEG	-	Neck node/0.9	Neck dissection and radiotherapy	NED/1.5
MI/M/48.1	II	1a	DEEP	Yes (no residual tumor at histology)	Larynx/1.2	CO_2_ TOLMS	NED/4.8
LM/M/68.7	II	1b	CD	No	Larynx/0.5	CO_2_ TOLMS	NED/5
PA/M/74.1	Va	1b	NEG	-	Larynx/0.8	CO_2_ TOLMS	NED/5
CA/M/70.3	III	1b	NEG	-	Larynx/0.6	CO_2_ TOLMS	DOC/2.2
MV/M/65.7	II	1b	CD	No	Larynx/1.3	CO_2_ TOLMS	NED/1.6

CO_2_ TOLMS = transoral CO_2_ laser microsurgery. NEG = negative. CD = close deep. DEEP = positive deep. NED = no evidence of disease. DOD = died of disease. DOC = died of other causes.

**Table 3 cancers-15-01490-t003:** Survival rates of the cohort of patients who underwent CO_2_ TOLMS for early glottic cancer.

	DSS 5 Years	RFS 5 Years	LCL 5 Years	OLP 5 Years
All patients (n = 351)	99.6%	92.9%	94.6%	98.2%
NEG margins (n = 286)	100%	93.2%	96.4%	98.7%
CS margin (n = 8)	100%	100%	100%	100%
CD margin (n = 15)	100%	86.7%	100%	100%
SS margin (n = 16)	100%	100%	100%	100%
MS margins (n = 9)	100%	100%	88.9%	100%
DEEP margin (n = 17)	92.9%	80.9%	57.5%	86.9%

DSS = disease-specific survival. RFS = recurrence-free survival. LCL = local control with laser alone. OLP = overall laryngeal preservation. NEG = negative. CS = close superficial. CD = close deep. SS = positive single superficial. MS = positive multiple superficial. DEEP = positive deep.

**Table 4 cancers-15-01490-t004:** Univariate analysis of different types of margins for recurrence-free survival.

	Recurrence	5-Year RFS	Hazard Ratio (95% CI)	*p*
NEG margins (n = 286)	17 (5.9%)	93.2%	1 (Reference)	NA
CD margin (n = 15)	2 (13.3%)	86.7%	2.537 (0.2854–22.55)	=0.2
DEEP margin (n = 17)	3 (17.6%)	80.9%	2.863 (0.4395–18.66)	=0.08

NA = not applicable. CI = confidence interval. NEG = negative. CD = close deep. DEEP = positive deep.

## Data Availability

The data that support the findings of this study are available as electronic Appendix A.

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
