# Peer review of "Management and Oncologic Outcomes of Close and Positive Margins after Transoral CO2 Laser Microsurgery for Early Glottic Carcinoma"

_cancers, 2023, doi:10.3390/cancers15051490_

Round 1

Reviewer 1 Report

Thank for let me review the pereset paper. The article present a very complete and well organized study on the role of close and positive margins in TLS. The peresnt study gives strong evidence regarding the optimal management of these patients.

Overall a very interesting work.

Only few questions:

1) Why do you routinely use a III generation cephalosporine (ceftriaxone) for antibiotic prophilaxis?

2) In discussion yopu rpeorted the data of close and positive margins after introducing NBI. Please report the data also in Result section.

Author Response

We found the reviewer’s comments very helpful, and we believe that his/her suggestions will contribute to improve the quality of this work.

  1. The reviewer asked the authors to explain why do we routinely use a III generation cephalosporine (ceftriaxone) for antibiotic prophylaxis.

Answer: The authors are aware that infections after CO2 TOLMS are extremely rare; however, the hospital protocol advises the administration of antibiotic prophylaxis with cephalosporine even for endoscopic procedures.

This point was made clearer in the Materials and Methods section of the revised version of the manuscript.

  1. The reviewer asked the authors to further detail the incidence of superficial margins after introducing NBI in the Results section of the manuscript.

Answer: According with his/her suggestion, the authors changed the Results section of the manuscript as follows:

“Seventeen out of the 229 patients (7.4%) who underwent CO2 TOLMS before the systematic intraoperative use of enhancement systems (NBI, IMAGE1 S and ECE), had positive superficial margins. After the implementation of bioendoscopy, only 8 of 122 patients (6.6%) experienced positive superficial margins. Therefore, the use of enhancement tools reduced the incidence of positive superficial margins, although the decrease was not statically significant (p=0.76).”

Reviewer 2 Report

Transoral laryngeal laser surgery is already an established method for the management of early glottic neoplasm. The size of the resection margins is important for independent prognosis in local control. Lately, the stratification of various types of resection margins has been of major importance from a prognostic perspective. The present study is of particular importance because it establishes a practical algorithm for the management of resection margins. This is an excellent paper for every clinician involved in the management of early gottic cancer.

Author Response

The authors thank the reviewer for his/her comments

Reviewer 3 Report

Thank you very much for this article. It emphasizes the importance of a well experienced surgeon coupled with a dedicated head and neck pathologist to fully understand resection margins in laser resection of early larynx cancers. I look forward to reading another article that will hopefully compare CIS, T1 and T2's and their margin results after laser surgery

Author Response

The authors thank the reviewer for his/her comments.